# Learning Semantic-Enhanced Dual Temporal Adjacent Maps for Video Moment Retrieval

## Abstract

Retrieving a specific moment from an untrimmed video via a text description is a central problem in vision-language learning. It is a challenging task due to the sophisticated temporal dependency among moments. Existing methods fail to deal with this issue well since they establish temporal relations of moments in a way that visual content and semantics are coupled. This paper studies temporal dependence schemes that decouple content and semantic information, establishing semantic-enhanced **D**ual **T**emporal **A**djacent **M**aps for video moment retrieval, conferred as DTAM. Specifically, DTAM designs two branches to encode visual appearance and semantic knowledge from video clips respectively, where knowledge from the appearance branch is distilled into the semantic branch to help DTAM distinguish features with the same visual content but different semantics with a well-designed semantic-aware contrastive loss. Besides, we also develop a moment-aware mechanism to assist temporal adjacent maps' learning for better video grounding. Finally, extensive experimental results and analysis demonstrate the superiority of the proposed DTAM over existing state-of-the-art approaches on three challenging video moment retrieval benchmarks, *i.e.*, TACoS, Charades-STA, and ActivityNet Captions.

## 1 Introduction

Retrieving moments in an untrimmed video based on text descriptions involves joint reasoning about visual, linguistic and temporal dependency. It therefore remains a challenging problem in video understanding. With the rapid development of vision-language learning, video moment retrieval has achieved significant advances, including applications in video question answering (Wang et al., 2024a; Xiao et al., 2024a), interactive video editing (Tu et al., 2024; Shi et al., 2024), and intelligent personal assistants (Nasri et al., 2023). Specifically, the video moment retrieval task aims to localize segments based on a given text query. For example, given a description "*A man pushes a child on the swing set of a large park area*" as shown in Fig. 1 (*Query A*) and a paired video, this task is asked to retrieve the best matching moment from the video.

Prevailing solutions for video moment retrieval have achieved remarkable advances and mainly fall into two groups, *i.e.,* two-stage (Zhang et al., 2023; Xiao et al., 2021; Xu et al., 2019; Chen & Jiang, 2019; Zhang et al., 2021b) and one-stage (Mu et al., 2024; Zeng et al., 2020; Lin et al., 2023; Lei et al., 2021; Zhang et al., 2019a). The former generates candidate moments by operating well-designed sliding windows on raw videos. Then each candidate moment is combined with the corresponding text description to determine whether they are matched. Apparently, such a pipeline is inspired by object detection techniques for images, but ignores the flexibility and complexity of moment description, resulting in an inability to encode temporal dependencies between moments that are crucial for moment retrieval. For example, as displayed in Fig. 1 (*Query C*) "*The man pushes the swing to get the child started **again***", the sentence query requires algorithms to realize that the action happened twice. As a result, two-stage models will generate false positives since each candidate moment is independent and the model only observes part of the video. To address this problem, one-stage approaches attempt to localize moments directly. Especially, some efforts (Zhang et al., 2020; Gao et al., 2021; Zhang et al., 2021a; Soldan et al., 2021; Wang et al., 2021) introduce temporal adjacent maps to represent candidate moments with their coordinates naturally. Such temporal adjacent networks can cover diverse moments with various lengths while expressing their adjacent relations, which is suitable for capturing temporal dependencies among moments. In this fashion,

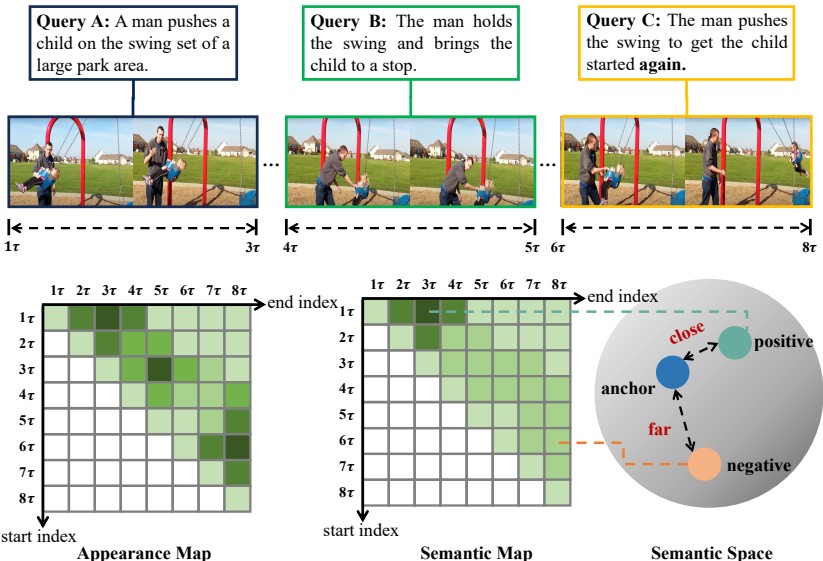

Figure 1: Illustration of our motivation and solutions for moment retrieval with referring expressions. We observe that interested video clips have similar visual appearance but different semantics (*i.e.*, corresponding to different natural language queries). Thus, the dual branches are employed to explicitly encode appearance and semantics in a decoupled manner, facilitating differentiation between instances with similar appearance but different semantics. Values in 2D maps represent the matching scores and the color brightness highlights the degree of matching. $\tau$ is a duration determined by the video length and sampling rate.

the model is capable of perceiving contextual information and semantic entailment. Nevertheless, these vanilla temporal adjacent maps encode visual appearance and semantic knowledge in a coupled manner, making it difficult to distinguish instances with similar content but different semantics. As shown in Fig. 1, the video clip corresponding to *Query B* looks similar in appearance to the video clip corresponding to *Query A*, but they express different meanings.

To this end, we propose a semantic-enhanced **D**ual **T**emporal **A**djacent **M**aps (DTAM) for effective video grounding. Specifically, we construct augmented appearance maps and semantic maps with a novel moment-aware mechanism to explicitly capture visual appearance and semantics, respectively. Prediction results from the appearance map are further utilized to provide rich semantic signals to assist the semantic map's learning. Besides, we argue that retrieved fragments corresponding to the same query should have similar semantics, and thus a semantic-aware contrastive loss is developed to encourage normalized embeddings corresponding to the same query to be pulled closer together, while embeddings corresponding to different queries are repelled apart.

In summary, the main contributions and innovations of this work are summarized as follows:

1. We propose a novel semantic-enhanced **D**ual **T**emporal **A**djacent **M**aps (DTAM) for effective video moment retrieval, which models temporal dependencies between moments in an appearance-semantic decoupled fashion.

2. We argue that retrieved fragments corresponding to the same query should have similar semantics, and therefore develop a semantic-aware contrastive loss to instruct the semantic map's learning with the weak signals provided by the appearance map.

3. A novel moment-aware mechanism is proposed to augment vanilla temporal adjacent maps so that they can perceive the importance of each moment.

4. The proposed DTAM achieves satisfactory performance on three challenging video moment retrieval benchmarks, *i.e.*, TACoS, Charades-STA, and ActivityNet Captions. Extensive ablation studies and analyses uncover the effectiveness of the proposed components.

## 2 RELATED WORK

### 2.1 TEMPORAL ACTION LOCALIZATION

Temporal action localization aims to localize and identify action instances in untrimmed videos. The representative paradigms include two-stage temporal detection pipelines (Lin et al., 2021; Shi et al., 2023; Yan et al., 2023; Xu et al., 2020) and one-stage single-shot methods (Wang et al., 2023; 2024c; Zhang et al., 2022; Liu et al., 2022c; Rizve et al., 2023). Specifically, the two-stage pipeline first generates candidate action instances along the temporal dimension, and then the boundaries of these candidate regions are further refined via well-designed regressors. Instead, the one-stage methods directly predict action probability scores of each video clip. After that, some well-designed aggregation operations are utilized to generate instance-level results. However, the action categories of this task are limited to a predefined set and fail to deal with complex and open-set actions in real scenarios. As a consequence, moment retrieval with natural language query (Anne Hendricks et al., 2017; Gao et al., 2017) is proposed to alleviate this problem.

### 2.2 VIDEO AND LANGUAGE CROSS-MODAL FUSION

Video and language cross-modal fusion is a fundamental problem for multi-modal video understanding, which plays an important role in many downstream tasks (Liu et al., 2022b; Chen et al., 2024; Wang et al., 2024b; Wang & Chen, 2021). Different from image-language fusion, video-language cross-modal fusion involves the complicated temporal context of video and semantic interactions between video representations and their corresponding referring expressions, which is a fairly intractable problem. Existing solutions mainly include sequential modeling (Chen et al., 2018; Zhang et al., 2019c) and attention-based strategy (Lei et al., 2021; Xu et al., 2023; Barrios et al., 2023; Mu et al., 2024; Lin et al., 2023; Soldan et al., 2021; Zhang et al., 2021a). For sequential modeling, the core idea is to temporally align video and text descriptions using recurrent neural networks. Chen *et al.* (Chen et al., 2018) propose this paradigm for the first time, and attempt to capture evolving fine-grained frame-by-word interactions between video and text. Zhang *et al.* (Zhang et al., 2019c) further investigate solutions based on a bidirectional GRU to align video-text features. For attention-based strategy, algorithms employ the attention mechanism to adaptively fuse video features with referring expressions. Zhang *et al.* (Zhang et al., 2019b) and Hendricks *et al.* (Hendricks et al., 2018) adopt a hard attention strategy to refine visual representations based on text descriptions. Instead, some efforts investigate the soft attention paradigm (Xu et al., 2023; Barrios et al., 2023; Mu et al., 2024; Lin et al., 2023; Soldan et al., 2021) for cross-modal representation fusion. For instance, MH-DETR (Xu et al., 2023) adopts a combination of self-attention and cross-attention to emphasize the aspects of the visual content most relevant to the textual semantics. Barrios *et al.* (Barrios et al., 2023) explore a multi-modal attention mechanism to guide feature fusion. In this paper, we utilize a simple multiplication operation for efficient and effective multi-modal feature fusion, instead of expensive attention paradigms or complicated temporal strategies.

### 2.3 VIDEO MOMENT RETRIEVAL

Existing popular methods of video moment retrieval can be roughly divided into two categories, *i.e.*, one-stage and two-stage. One-stage methods directly retrieve moments without proposal generation. Some algorithms predict the onset and offset probabilities (Liu et al., 2021b; Nan et al., 2021; Zhao et al., 2021; Zhang & Radke, 2022) or optimize anchors (Wang et al., 2020; Yuan et al., 2022; Zhang et al., 2019a) for each moment. Another line of work utilizes learnable semantic queries (Lin et al., 2023; Lei et al., 2021; Sun et al., 2024) and global representations (Li et al., 2022; Zhou et al., 2021; Li et al., 2021) to infer accurate moment boundaries. Inspired by point-based object detection (Tian et al., 2022), some works treat moment proposals as points (Mu et al., 2024; Zeng et al., 2020; Fang et al., 2023; Liu et al., 2022a; Chen et al., 2020). The moment boundaries are regressed relative to these points. Although efficient, the retrieval performance of one-stage methods is unsatisfactory. In contrast, two-stage approaches first generate candidate segment proposals, then assess their "actionness" and optionally refine their temporal boundary. These researches (Anne Hendricks et al., 2017; Gao et al., 2017; Yang et al., 2024) sample candidate proposals using dense sliding windows and evaluate them independently. To capture the dynamics of videos, temporal feature modeling is studied, such as adopting RNN (Rodriguez et al., 2020; Yuan et al., 2019) and graphs (Sun et al., 2023; Zheng et al., 2023) to explore the dynamic relations of videos. Recently, some efforts (Ma

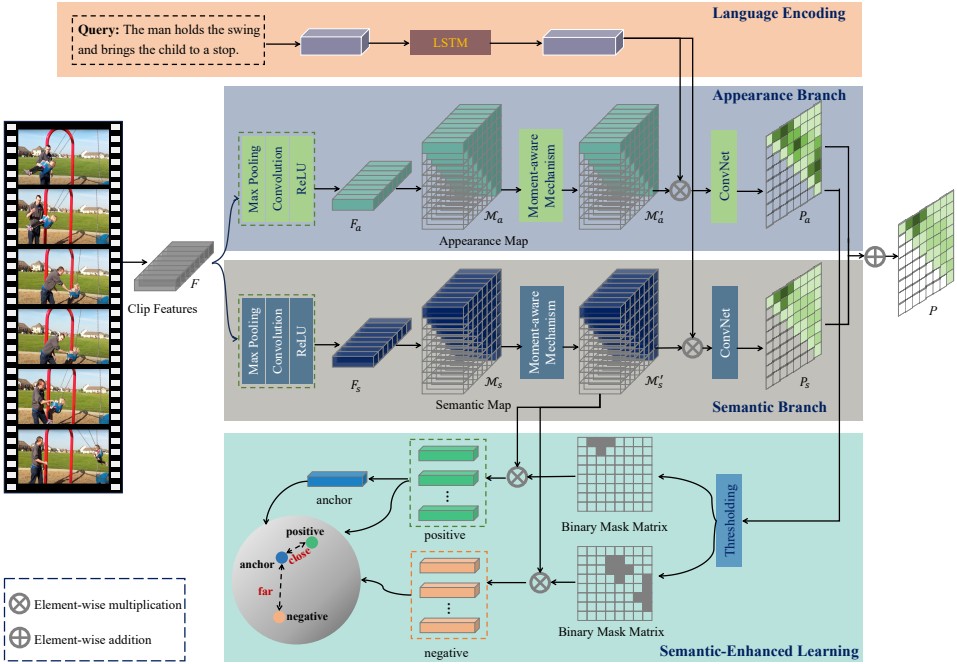

Figure 2: The overview of DTAM. The proposed DTAM utilizes a text encoder to extract language representations, and establishes a dual temporal adjacent map to capture visual appearance and semantics from video clips explicitly. Moreover, knowledge from the appearance branch is distilled into the semantic branch to help DTAM distinguish features with the same visual content but different semantics with a well-designed semantic-aware mechanism.

et al., 2023; Luo et al., 2023; Huang et al., 2024; Xiao et al., 2024b) also endeavored to empower large language models to grasp video moments.

To reduce computational overhead, some efforts (Xiao et al., 2021; Xu et al., 2019; Chen & Jiang, 2019) attempt to alleviate dense sampling by using textual descriptions as guidance information. Towards the opposite optimization direction, some approaches based on temporal adjacent maps (Zhang et al., 2020; Gao et al., 2021; Zhang et al., 2021a; Soldan et al., 2021; Wang et al., 2021) instead enumerate all possible proposals, focusing on capturing underlying temporal dependencies between moments and achieving satisfactory performance. However, they encode visual appearance and semantic knowledge in a coupled temporal adjacent map, distinguishing instances with similar content but different semantics difficult. To this end, we propose a dual temporal adjacent map to encode visual appearance and semantics in a decoupled fashion, facilitating more efficient video moment retrieval.

## 3 METHODOLOGY

This section first presents the formulation of video moment retrieval and recaps the paradigm of vanilla temporal adjacent maps, then demonstrates an overview of the proposed DTAM framework and detailed principles of each component. Finally, extensive experiments, ablation analysis, and visualization results are conducted on three challenging video moment retrieval benchmarks, *i.e.*, TACoS (Regneri et al., 2013), Charades-STA (Sigurdsson et al., 2016), and ActivityNet Captions (Krishna et al., 2017).

### 3.1 PROBLEM FORMULATION

Given an untrimmed video $V$ and a referring expression $S$ as a query, video moment retrieval aims to find the video segments that best match the referring expression. Formally, an input video $V$ contains $l_v$ frames, *i.e.*, $V = \{x_i\}_{i=0}^{l_v-1}$, where $x_i$ represents a frame in the video. The referring expression $S$ is a text description, *i.e.*, $S = \{s_i\}_{i=0}^{l_s-1}$, where $s_i$ represents a word and $l_s$ is the number of words in

the sentence. Video moment retrieval is expected to localize consecutive frames from $x_i$ to $x_j$ with the same semantics as the query $S$.

## 3.2 Recap of Temporal Adjacent Maps

Temporal Adjacent Map is pioneered by (Zhang et al., 2020) to encode temporal dependencies between moments for video moment retrieval. Given an untrimmed video as input, it is segmented into non-overlapped video clips, each represented as $v_i$ containing $T$ consecutive frames. Next, a fixed-interval sampling over these video clips is conducted, resulting in $N$ video clips $V = \{v_i\}_{i=0}^{N-1}$. For each clip, a visual feature extractor followed by a fully connected layer is employed to acquire a compact representation $F \in \mathcal{R}^d$. These sampled clips $\{F_i\}_{i=0}^{N-1}$ are further used to generate candidate moments with a sparse sampling strategy, whose core idea is to apply dense sampling on moments of short duration and increasing sampling intervals on longer duration. Specifically, all possible moments are chosen as candidates when $N \leq 16$. When $N > 16$, a moment starting from clip $v_a$ to $v_b$ is chosen as the candidate when satisfying the following constraint:

$$(a \bmod s = 0) \,\&\, ((b - s') \bmod s = 0), \tag{1}$$

where $a$ and $b$ are the indexes of clips, $s$ and $s'$ are defined as :

$$s = 2^{k-1},$$
$$s' = \begin{cases} 0, & if \quad k = 1 \\ 2^{k+2} - 1, & otherwise \end{cases} \tag{2}$$

where $k = \lceil \log_2(\frac{b-a+1}{8}) \rceil$ and $\lceil \cdot \rceil$ is a ceiling operation. Through this sparse sampling strategy, we can cover various moments with different lengths while reducing computational overhead and redundant candidates (Zhang et al., 2020; 2021b).

Then, these candidate moments are utilized to construct temporal adjacent maps. Specifically, for each candidate moment, a max-pooling operation is applied on video clip features across a specified time span, generating the representation of the moment:

$$F_{a,b} = maxpool(F_a, ..., F_b), \tag{3}$$

where $a$ and $b$ are the start and end indices of clips respectively.

Based on all these candidate moment representations, a 2D temporal adjacent map $\mathcal{M} \in \mathcal{R}^{N \times N \times d}$ is constructed, where the first two dimensions represent the start and end indices of clips respectively, and the last one denotes the representation dimension. For example, $\mathcal{M}[a, b, :]$ represents a moment from clip $v_a$ to $v_b$. Notably, a legal index range needs to be guaranteed, *i.e.*, $a \leq b$, while values in illegal intervals are padded with zeros. According to candidate indicators in Eq. 1, all valid scores on the map are gathered, where the moment candidate with the highest score is regarded as the retrieved result. Intuitively, such a 2D temporal adjacent map covers various candidate moments with different lengths while encoding temporal relations of moments, which is an attractive component for the video moment retrieval task.

## 3.3 Overview

In this section, we develop a semantic-enhanced dual temporal adjacent map for video moment retrieval, as demonstrated in Fig. 2, where an appearance map and a semantic map are constructed to capture visual appearance and moment semantics respectively. Specifically, the representation $F_q \in \mathcal{R}^d$ of the referring expression $S$ is extracted through a GloVe algorithm (Pennington et al., 2014) followed by a bidirectional LSTM network. For the video input, we extract representations $F_a$ and $F_s$ for the appearance branch and semantic branch respectively by applying a MaxPooling-Convolution-ReLU operation on sampled clips $F$. They then are leveraged to generate an appearance temporal adjacent map $\mathcal{M}_a \in \mathcal{R}^{N \times N \times d}$ and a semantic temporal adjacent map $\mathcal{M}_s \in \mathcal{R}^{N \times N \times d}$ with the procedure described in Section 3.2. To enhance the capabilities of perceiving moments, a novel moment-aware mechanism, as described in Section 3.5, is proposed to refine the discriminability of $\mathcal{M}_a$ and $\mathcal{M}_s$, leading to $\mathcal{M}'_a$ and $\mathcal{M}'_s$. Then we conduct a multi-modal representation fusion over $\mathcal{M}'_a$, $\mathcal{M}'_s$ and $F_q$ respectively through a simple matrix operation:

$$\widetilde{\mathcal{M}'_a} = ||\mathcal{M}'_a \otimes (\mathcal{I} \odot (F_q)^T)||_F,$$
$$\widetilde{\mathcal{M}'_s} = ||\mathcal{M}'_s \otimes (\mathcal{I} \odot (F_q)^T)||_F, \tag{4}$$

where $\mathcal{I} \in \mathcal{R}^{N \times N \times 1}$ is an all-ones tensor, $\otimes$ denotes element-wise multiplication, $\odot$ denotes matrix multiplication, and $|| \cdot ||_F$ denotes Frobenius normalization. Subsequently, we further apply ConvNets containing $L$ convolutional layers with the kernel size of $K$ on $\widetilde{\mathcal{M}'_a}$ and $\widetilde{\mathcal{M}'_s}$, followed a sigmoid function to generate an appearance score map $P_a \in \mathcal{R}^{N \times N \times 1}$ and a semantic score map $P_s \in \mathcal{R}^{N \times N \times 1}$. The final predictions are acquired through an average operation on them, *i.e.,* $P = \frac{P_a + P_s}{2}$. Each value in this final score map $P$ indicates the degree of confidence in matching the referring expression and candidate moment. To supervise this training process, we employ a scaled Intersection over Union (IoU) value to provide learning signals (Zhang et al., 2020). Especially, we first calculate IoU $u_i$ between each candidate moment and their corresponding ground truth. $u_i$ then is scaled by two thresholds $t_{max}$ and $t_{min}$ as follows:

$$
y_i = \begin{cases} 0, & u_i \leq t_{min}, \\ \frac{u_i - t_{min}}{t_{max} - t_{min}}, & t_{min} < u_i < t_{max}, \\ 1, & u_i \geq t_{max}. \end{cases} \tag{5}
$$

Here, $y_i$ is utilized as ground truth to supervise the training of DTAM, and the cross entropy loss is adopted on $P_a$, $P_s$, and $P$ as the learning criteria:

$$
\mathcal{L}_{score} = \frac{1}{M} \sum_{i=1}^{M} (y_i \log p_a^i + (1 - y_i) \log(1 - p_a^i) + y_i \log p_s^i + (1 - y_i) \log(1 - p_s^i)
$$
$$
+ y_i \log p^i + (1 - y_i) \log(1 - p^i)), \tag{6}
$$

where $p_a^i$, $p_s^i$, and $p^i$ are the candidates' output scores, whose values are from the score map $P_a$, $P_s$, and $P$ respectively, and $M$ is the number of valid moment candidates.

Besides, the semantic branch absorbs semantic knowledge from the appearance branch. Specifically, predictions from the appearance branch are distilled into the semantic branch to help DTAM distinguish features with the same visual content but different semantics with a well-designed semantic-aware contrastive loss, as described in Section 3.4.

## 3.4 LEARNING SEMANTIC TEMPORAL ADJACENT MAPS

In this paper, we argue that the retrieved fragments corresponding to the same query should have similar semantics. To achieve this goal, a semantic-aware contrastive loss is developed to encourage normalized embeddings of clips corresponding to the same query to be pulled closer together, while embeddings corresponding to different queries are repelled apart. Formally, the knowledge from the appearance branch is distilled to instruct the semantic branch's learning. For a given video, valid moments with a score from the $P_a$ higher than a pre-defined threshold $\sigma_a$ are regarded as positive samples, while moments with a score lower than a threshold $\sigma_b = \frac{1}{2}(1 - \sigma_a)$ are regarded as negative samples. Besides, the center of these positive samples serves as an anchor. Then, we apply the InfoNCE loss on the normalized embedding anchors and positives against the embedding of negative samples:

$$
\mathcal{L}_{nce} = -\frac{1}{M} \sum_{i=1}^{M} \log \frac{\exp\left(G_a^T G_p / \varrho\right)}{\sum_{j=1}^{M} \exp\left((G_a^T G_j / \varrho)\right)}, \tag{7}
$$

where $M$ is the number of valid moments, $\varrho$ is the temperature parameter that is set as 0.1, $G_a$, and $G_p$ denote the representation of anchor and positive sample, respectively.

The core of this paradigm is to obtain reliable positive and negative samples, which depends on the hyper-parameter $\sigma_a$ and the appearance branch's outputs. Thus, we first set a larger value for $\sigma_a$ to ensure reliability, and then investigate four different strategies on them, including (1) joint training for two branches and using a fixed $\sigma_a$; (2) joint training for two branches and using a cosine decline for $\sigma_a$; (3) joint training for two branches and using a linear decline for $\sigma_a$; and (4) pre-trained appearance branch and using a fixed $\sigma_a$. We explore their utilities via experiments in Section 4.2.

| Model | R@1 tIoU=0.3 | R@1 tIoU=0.5 | R@5 tIoU=0.3 | R@5 tIoU=0.5 |
|---|---|---|---|---|
| CTRL (Gao et al., 2017) | 18.32 | 13.30 | 36.69 | 25.42 |
| CMIN (Zhang et al., 2019c) | 24.64 | 18.05 | - | - |
| CBP (Wang et al., 2020) | 27.31 | 24.79 | 43.64 | 37.40 |
| SCDM (Yuan et al., 2022) | 26.11 | 21.17 | 40.16 | 32.18 |
| DRN (Zeng et al., 2020) | - | 23.17 | - | 33.36 |
| BPNet (Xiao et al., 2021) | 25.96 | 20.96 | - | - |
| CSMGAN (Liu et al., 2020) | 33.90 | 27.09 | 53.98 | 41.22 |
| 2D-TAN (Zhang et al., 2020) | 37.29 | 25.32 | 57.81 | 45.04 |
| CBLN (Liu et al., 2021b) | 38.98 | 27.65 | 59.96 | 46.24 |
| MMN (Wang et al., 2022) | 39.24 | 26.17 | 62.03 | 47.39 |
| GTR (Cao et al., 2021) | 40.39 | 30.22 | 61.94 | 47.73 |
| MS-2D-TAN (Zhang et al., 2021b) | 45.61 | 35.77 | 69.11 | 57.31 |
| IVG-DCL (Nan et al., 2021) | 38.84 | 29.07 | - | - |
| IA-Net (Liu et al., 2021c) | 37.91 | 26.27 | 57.62 | 46.39 |
| APGN (Liu et al., 2021a) | 40.47 | 27.86 | 59.98 | 47.12 |
| FVMR (Gao & Xu, 2021) | 41.48 | 29.12 | 64.53 | 50.00 |
| CPNet (Li et al., 2021) | 42.61 | 28.29 | - | - |
| SSCS (Ding et al., 2021) | 41.33 | 29.56 | 60.65 | 48.01 |
| RaNet (Gao et al., 2021) | 43.34 | 33.54 | 67.33 | 55.09 |
| MGSL-Net (Liu et al., 2022a) | 42.54 | 32.27 | 63.39 | 50.13 |
| MATN (Zhang et al., 2021a) | 48.79 | 37.57 | 67.63 | 57.91 |
| VLG-Net (Soldan et al., 2021) | 45.46 | 34.19 | 70.38 | 56.56 |
| SSRN (Zhu et al., 2023) | 45.10 | 34.33 | 65.26 | 51.85 |
| G2L (Li et al., 2023) | 42.74 | 30.95 | 65.83 | 49.86 |
| MomentDiff (Li et al., 2024) | 44.78 | 33.68 | - | - |
| **DTAM(Ours)** | **49.65** | **39.72** | **71.83** | **58.96** |

Table 1: Retrieval performance comparisons on the TACoS dataset using C3D features.

| Model | R@1 tIoU=0.5 | R@1 tIoU=0.7 | R@5 tIoU=0.5 | R@5 tIoU=0.7 |
|---|---|---|---|---|
| CTRL (Gao et al., 2017) | 29.01 | 10.34 | 59.17 | 37.54 |
| QSPN (Xu et al., 2019) | 27.70 | 13.60 | 59.20 | 38.30 |
| ACRN (Liu et al., 2018) | 31.67 | 11.25 | 60.34 | 38.57 |
| SCDM (Yuan et al., 2022) | 36.90 | 20.28 | 66.84 | 42.92 |
| TripNet (Hahn et al., 2020) | 32.19 | 13.93 | - | - |
| CBP (Wang et al., 2020) | 35.76 | 17.80 | 65.89 | 46.20 |
| LGI (Mun et al., 2020) | 41.51 | 23.07 | - | - |
| 2D-TAN (Zhang et al., 2020) | 44.51 | 26.54 | 77.13 | 61.96 |
| DRN (Zeng et al., 2020) | 45.45 | 24.36 | 77.97 | 50.30 |
| BPNet (Xiao et al., 2021) | 42.07 | 24.69 | - | - |
| CPNet (Li et al., 2021) | 40.56 | 21.63 | - | - |
| SMIN (Wang et al., 2021) | 48.46 | 30.34 | 81.16 | 62.11 |
| CBLN (Liu et al., 2021b) | 48.12 | 27.60 | 79.32 | 63.41 |
| CPN (Zhao et al., 2021) | 45.10 | 28.10 | - | - |
| DeNet (Zhou et al., 2021) | 43.79 | - | 74.13 | - |
| MATN (Zhang et al., 2021a) | 48.02 | 31.78 | 78.02 | 63.18 |
| VLG-Net (Soldan et al., 2021) | 46.32 | 29.82 | 77.15 | 63.33 |
| APGN (Liu et al., 2021a) | 48.92 | 28.64 | 78.87 | 63.19 |
| IA-Net (Liu et al., 2021c) | 48.57 | 27.95 | 78.99 | 63.12 |
| RaNet (Gao et al., 2021) | 45.59 | 28.67 | 75.93 | 62.97 |
| MGSL-Net (Liu et al., 2022a) | 51.87 | 31.42 | 82.60 | 66.71 |
| MMN (Wang et al., 2022) | 48.59 | 29.26 | 79.50 | 64.76 |
| G2L (Li et al., 2023) | 51.68 | 33.35 | 81.32 | 67.60 |
| SnAG (Mu et al., 2024) | 48.55 | 30.56 | 81.71 | 63.41 |
| **DTAM(Ours)** | **52.44** | **33.98** | **83.70** | **67.91** |

Table 2: Retrieval performance comparisons on the ActivityNet-Captions using C3D features.

### 3.5 MOMENT-AWARE MECHANISM

To enhance the semantics of temporal adjacent maps, we propose a novel moment-aware mechanism to emphasize the importance of each moment, whose visualization can be found in Appendix A.1. Given a temporal adjacent map $\mathcal{M} \in \mathcal{R}^{N \times N \times d}$, each element $\mathcal{M}_{ij}$ indicates a candidate moment starting from the clip $v_i$ and ending at $v_j$. The core idea is to evaluate each clip's importance as the beginning and end of a moment. Specifically, we squeeze the values of $\mathcal{M}$ along the first dimension with an average operation and acquire $Q_s \in \mathcal{R}^{1 \times N \times d}$, representing the likelihood that each clip is the *start* of a moment. Similarly, we squeeze the values of $\mathcal{M}$ along the second dimension with an average operation and acquire $Q_e \in \mathcal{R}^{N \times 1 \times d}$, representing the likelihood that each clip is the *end* of a moment. Then, $Q_s$ and $Q_e$ are fed into a squeeze module of "Conv-ReLU-Conv-Sigmoid", leading to the attention scores $Q_s' \in \mathcal{R}^{1 \times N \times 1}$ and $Q_e' \in \mathcal{R}^{N \times 1 \times 1}$. Afterwards, they are applied on the original $\mathcal{M}$ to yield the refined temporal adjacent map as follows:

$$\mathcal{M}' = Q_s' \otimes \mathcal{M} + Q_e' \otimes \mathcal{M}, \tag{8}$$

where $\otimes$ indicates element-wise multiplication with a broadcast mechanism in Python. In this fashion, we learn explicitly semantic-enhanced temporal adjacent maps by perceiving moment importance, and the effectiveness is also verified and analyzed in experiments.

### 3.6 OVERALL LOSS FUNCTION

To optimize the proposed DTAM during training, the overall loss function is formulated as:

$$\mathcal{L}_{all} = \mathcal{L}_{score} + \lambda \mathcal{L}_{nce}, \tag{9}$$

where $\lambda$ is a hyper-parameter that controls the importance of different terms.

## 4 EXPERIMENTS

In this section, we conduct adequate comparisons of DTAM with prevailing baselines on three challenging video moment retrieval benchmarks. Then, we also provide comprehensive ablation studies to verify the utilities of each component of DTAM. Next, we will elaborate on the details of datasets, experimental settings, and experimental results and analysis.

### 4.1 DATASET AND SETTING

**Charades-STA Dataset**. Charades-STA (Sigurdsson et al., 2016) consists of 9848 videos of daily indoor activities, where each video has an average duration of 30 seconds and ∼2.4 text queries. 12408 moment-description pairs are used for training and 3720 pairs for testing.

| Model | Feature | R@1 tIoU=0.5 | R@1 tIoU=0.7 | R@5 tIoU=0.5 | R@5 tIoU=0.7 |
|---|---|---|---|---|---|
| SCDM (Yuan et al., 2022) | I3D | 54.92 | 34.26 | 76.50 | 60.02 |
| 2D-TAN (Zhang et al., 2020) | I3D | 56.64 | 36.21 | 89.14 | 61.13 |
| DRN (Zeng et al., 2020) | I3D | 53.09 | 31.75 | 89.06 | 60.05 |
| BPNet (Xiao et al., 2021) | I3D | 50.75 | 31.64 | - | - |
| MS-2D-TAN (Zhang et al., 2021b) | I3D | 56.64 | 36.21 | 89.14 | 61.13 |
| CPNet (Li et al., 2021) | I3D | 60.27 | 38.74 | - | - |
| SMIN (Wang et al., 2021) | I3D | 64.06 | 40.75 | 89.49 | 68.09 |
| CBLN (Liu et al., 2021b) | I3D | 61.13 | 38.22 | 90.33 | 61.69 |
| CPN (Zhao et al., 2021) | I3D | 51.07 | 31.54 | - | - |
| DeNet (Zhou et al., 2021) | I3D | 59.70 | 38.52 | 91.24 | 66.83 |
| APGN (Liu et al., 2021a) | I3D | 62.58 | 38.86 | 91.24 | 62.11 |
| IA-Net (Liu et al., 2021c) | I3D | 61.29 | 37.91 | 89.78 | 62.04 |
| RaNet (Gao et al., 2021) | I3D | 43.34 | 33.54 | 67.33 | 55.09 |
| MGSL-Net (Liu et al., 2022a) | I3D | 63.98 | 41.03 | 93.21 | 63.85 |
| SSRN (Zhu et al., 2023) | I3D | 65.59 | 42.65 | **94.76** | 65.48 |
| QD-DETR (Moon et al., 2023) | SlowFast+CLIP | 57.31 | 32.55 | - | - |
| LLaViLo (Ma et al., 2023) | LLaMa | 55.72 | 33.43 | - | - |
| VTimeLLM (Huang et al., 2024) | Vicuna | 34.30 | 14.70 | - | - |
| UVCOM (Xiao et al., 2024b) | SlowFast | 59.25 | 36.64 | - | - |
| MomentDiff (Li et al., 2024) | SlowFast+CLIP | 55.57 | 32.42 | - | - |
| **DTAM(Ours)** | I3D | **65.93** | **43.14** | 92.86 | **66.17** |

Table 3: Retrieval performance comparisons on Charades-STA using I3D features. The best results are in bold and second best underlined.

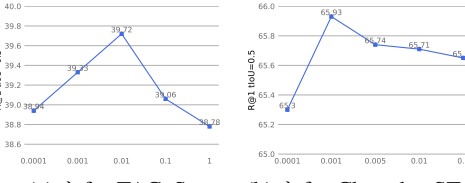

(a) $\lambda$ for TACoS  (b) $\lambda$ for Charades-STA

(c) $\lambda$ for ActivityNet Caption  (d) Explorations on distillation strategies on TACoS

Figure 3: Effects of hyper-parameters and distillation strategies.

| Model | R@1 tIoU=0.3 | R@1 tIoU=0.5 | R@5 tIoU=0.3 | R@5 tIoU=0.5 |
|---|---|---|---|---|
| APB | 45.14 | 32.07 | 65.61 | 53.46 |
| APB+MAM | 46.96 | 34.12 | 68.02 | 55.77 |
| APB+SEB | 48.14 | 37.88 | 69.91 | 57.56 |
| APB+SEB+MAM | **49.65** | **39.72** | **71.83** | **58.96** |

(a). Ablation study on model structure for TACoS.

| Model | R@1 tIoU=0.5 | R@1 tIoU=0.7 | R@5 tIoU=0.5 | R@5 tIoU=0.7 |
|---|---|---|---|---|
| APB | 47.01 | 28.93 | 78.98 | 62.67 |
| APB+MAM | 48.81 | 30.06 | 80.22 | 64.59 |
| APB+SEB | 50.99 | 32.07 | 81.89 | 66.05 |
| APB+SEB+MAM | **52.44** | **33.98** | **83.70** | **67.91** |

(b). Ablation study on model structure for ActivityNet Captions

| Model | R@1 tIoU=0.5 | R@1 tIoU=0.7 | R@5 tIoU=0.5 | R@5 tIoU=0.7 |
|---|---|---|---|---|
| APB | 58.93 | 38.55 | 89.36 | 62.87 |
| APB+MAM | 60.76 | 40.84 | 90.27 | 64.64 |
| APB+SEB | 62.91 | 42.00 | 91.68 | 65.39 |
| APB+SEB+MAM | **65.93** | **43.14** | **92.86** | **66.17** |

(c). Ablation study on model structure for Charades-STA.

| Model | Training ↓ | Inference ↓ | R@1 tIoU=0.5 ↑ |
|---|---|---|---|
| 2D-TAN (Zhang et al., 2020) | 63ms | 23ms | 44.51 |
| MS-2D-TAN (Zhang et al., 2021b) | 110ms | 35ms | 46.16 |
| APB | 47ms | 20ms | 47.01 |
| APB+MAM | 49ms | 22ms | 48.81 |
| APB+SEB | 71ms | 25ms | 50.99 |
| APB+SEB+MAM | 74ms | 31ms | 52.44 |

(d). Comparisons on training and inference speed on the ActivityNet Captions dataset. We report the average time taken for a single video on a Nvidia GeForce RTX4090 GPU.

Table 4: Ablation study on DTAM. "APB" indicates the appearance branch, "SEB" denotes the semantic branch, and "MAM" denotes the moment-aware mechanism.

**ActivityNet Captions Dataset**. ActivityNet Captions (Krishna et al., 2017) is a larger-scale video moment retrieval dataset, which consists of 19209 videos involving diverse scenes and contents. Each video has an average duration of two minutes and ∼3.6 text queries. We follow (Zhang et al., 2019c) and employ 37417 moment-description pairs for training and 17031 pairs for testing.

**TACoS Dataset**. TACoS (Regneri et al., 2013) is a conventional video moment retrieval dataset consisting of 127 videos of cooking activities taking place in the kitchen, where 10146, 4589 and 4083 text-video pairs are adopted for training, validation and testing, respectively.

**Evaluation Criteria**. We employ the same setting as (Gao et al., 2017) and report Rank@$k$ at various temporal intersection-over-union thresholds $\theta$ (R@$k$, tIoU=$\theta$) for all datasets. It measures the percentage of text queries with at least one of the top-$k$ moment predictions whose temporal overlap with the ground-truth moment is larger than $\theta$.

**Implementation Details**. For fair comparisons with previous work, we adopt I3D (Carreira & Zisserman, 2017) features as visual representations for Charades-STA, and C3D (Tran et al., 2015) features as visual representations for ActivityNet Captions and TACoS. LSTM is applied to extract language representations. The size of all hidden states is 512 in DTAM. Besides, the number of sampled video clips is set to 16 for Charades-STA, 64 for ActivityNet Captions, and 128 for TACoS. For the ConvNet block, $L = 8$ and $K = 3$ are adopted for Charades-STA and TACoS, and $L = 4$ and

$K = 9$ are adopted for ActivityNet Captions. The scaling thresholds $t_{min}$ and $t_{max}$ are set to 0.1 and 0.7 for TACoS and Charades-STA, and 0.5 and 1.0 for ActivityNet Captions. We adopt an Adam optimizer to train the proposed DTAM using a learning rate of 0.0001 with a warmup strategy, as well as an exponential moving average mechanism. The batch size is 128 for ActivityNet Captions and 32 for Charades-STA and TACoS. The non-maximum suppression strategy with a threshold of 0.5 is employed during inference.

## 4.2 Comparison with State-of-the-Art Methods

In this section, we conduct extensive experiments on three challenging video moment retrieval benchmarks, *i.e.,* Charades-STA, ActivityNet Captions, and TACoS, and report R@$k$ with the tIoU threshold $\theta$. Specifically, $k \in \{1, 5\}$ and $\theta \in \{0.3, 0.5\}$ are adopted for TACoS and $k \in \{1, 5\}$ and $\theta \in \{0.5, 0.7\}$ are adopted for Charades-STA and ActivityNet Captions. We compare the retrieval performance of the proposed DTAM with state-of-the-art methods under the same settings.

Comparisons with state-of-the-art methods on the TACoS dataset are shown in Table 1. Benefiting from the proposed dual temporal adjacent maps and semantic-enhanced mechanism, we observe that DTAM achieves the best retrieval performance on all evaluation criteria. Compared to previous best practices, DTAM achieves a 1.8% R@1 (against MATN (Zhang et al., 2021a)) and 2.1% R@5 (against VLG-Net (Soldan et al., 2021)) improvements at tIoU=0.3. It also acquires 5.7% R@1 (against MATN (Zhang et al., 2021a)) and 1.8% R@5 (against MATN (Zhang et al., 2021a)) improvements at tIoU=0.5. Besides, we also conduct experiments on a more challenging ActivityNet-Caption benchmark, and results are displayed in Table 2. Compared with state-of-the-art approaches, DTAM also achieves the best retrieval performance on all evaluation criteria. Compared to previous best practices, DTAM achieves a 1.1% R@1 (against MGSL-Net (Liu et al., 2022a)) and 1.3% R@5 (against MGSL-Net (Liu et al., 2022a)) improvements at tIoU=0.5. It also acquires 1.9% R@1 (against G2L (Li et al., 2023)) and 0.5% R@5 (against G2L (Li et al., 2023)) improvements at tIoU=0.7. Furthermore, comparisons with state-of-the-art approaches on the Charades-STA dataset are displayed in Table 3, and the proposed DTAM also achieves superior performance. Specifically, DTAM acquires 0.34 (R@1 at tIoU=0.5), 0.49 (R@1 at tIoU=0.7), and 0.69 (R@5 at tIoU=0.7) absolute percentage points higher than the latest SSRN (Zhu et al., 2023) respectively. Our DTAM is slightly worse than SSRN against R@1 at tIoU=0.5. The reason is that SSRN further performs temporal modeling on the original C3D features of videos. This is not the focus of this article and our focus is on the multi-modal vision-language matching ability of the model.

We also investigate the influence of hyper-parameters involved in the loss function, *i.e.*, $\lambda$, and results for R@1 at tIoU=0.5 are reported in Fig. 3. We can observe that $\lambda = 0.01$, $\lambda = 0.001$, and $\lambda = 0.005$ are the best choice for TACoS, Charades-STA, and ActivityNet Caption respectively, which are employed as default values for all other experiments in this paper. To evaluate DTAM's computational efficiency, the training and inference speed on the ActivityNet Caption dataset is tested on a Nvidia Geforce RTX4090 GPU, where the classic algorithms 2D-TAN (Zhang et al., 2020) and MS-2D-TAN (Zhang et al., 2021b) using 2D temporal adjacent maps are compared as baselines. From Table 4 (d), we observe that the complete DTAM outperforms the complex MS-2D-TAN in both speed and prediction accuracy. At the same time, it only takes almost negligible time more than 2D-TAN, but it achieves better retrieval accuracy by a large margin.

Since the core motivation of this paper is learning semantic knowledge, we explore four instruction strategies from the appearance branch into a semantic branch, where the threshold $\sigma_a$ is restricted to vary from 0.8 to 0.5, results are shown in Fig. 3 (d). We observe that a dynamic adjustment of $\sigma_a$ is better than using a fixed value and also the higher threshold is superior to the lower one in the initial stage (version with a cosine decline is better than linear decline), which is in line with our understanding. In the early stages of training, the outputs of model are unreliable, so a higher threshold is required. However, as the model's capabilities improve, the threshold can be relaxed. Another interesting phenomenon is that the dynamic $\sigma_a$ with a cosine decline is slightly superior to using a pre-trained appearance branch with a fixed $\sigma_a$, which reveals that dynamic joint learning of two branches can complement each other.

## 4.3 Ablation Study

In this section, we conduct extensive ablation studies on all datasets to evaluate the utilities of each component in DTAM and guarantee that the experimental results are convincing. Overall, quan-

Figure 4: Qualitative visualization of three examples with different model variants are illustrated. (a) An example contains various scenes, accompanied by rapid shot switching. (b) An example shows that all frames have similar visual appearance but different semantic information. (c) A failed example that requires the algorithm to have abilities in counting.

titative results about retrieval accuracy and computational consumption are reported, and the visualization analyses are also demonstrated to provide a comprehensive understanding of DTAM. Experimental results of retrieval accuracy are detailed in Table 4 (a) for TACoS, in Table 4 (b) for ActivityNet Caption, and in Table 4 (c) for Charades-STA, respectively. Also, significant performance improvements are observed whether the proposed semantic adjacent map or moment-aware mechanism is employed for all datasets, which proves that these novel mechanisms are beneficial for more precise retrieval. Interestingly, we also find that introducing the semantic adjacent map brings more gains than the moment-aware mechanism, which illustrates the importance of semantic information for video moment retrieval.

Besides, the computational consumption of different structure variants is also analyzed, as shown in Table 4 (d). The complete DATM only takes almost negligible time more than 2D-TAN, but it achieves better retrieval accuracy by a large margin. Besides, the dual structure (APB + SEB) takes about 1.6 times longer than the variant using only the appearance branch (APB) during training, but spends negligible additional overhead during inference. The reason is that the appearance branch and semantic branch are parallelized without guidance signal propagation between them.

## 4.4 QUALITATIVE VISUALIZATION

To demonstrate our method's video moment retrieval performance, we provide some examples with different model variants and visualize their prediction results, as illustrated in Fig 4. Overall, the complete DTAM achieves the best predictions against other counterparts. For the first example, as shown in Fig. 4 (a), the video contains content of different scenes, accompanied by rapid switching of shots, which requires the algorithm to have the ability of temporal dynamics modeling and fine-grained feature extraction. As observed from the visualization, we find that the proposed moment-aware mechanism and semantic adjacent map can effectively improve the accuracy of moment retrieval. Besides, as illustrated in Fig. 4 (b), the second example is that all frames have similar visual appearance but different semantic information. We notice that the localization performance with the semantic map has a significant boost, further validating the motivation of this paper. Moreover, we also give a failed example, as illustrated in Fig. 4 (c), which has almost the same content and is required to retrieve the *fourth* rubbing action on the patient's arm. This requires the algorithm to have the ability to count, which is intractable for current solutions that attempt to find the moment with the highest confidence, regardless of the frequency.

## 5 CONCLUSION

In this paper, we proposed to learn semantic-enhanced dual temporal adjacent maps *i.e.*, DTAM, for efficient video moment retrieval. In this paradigm, we investigate temporal dependency schemes that decouple the visual appearance and semantics of moments. Moreover, a novel moment-aware mechanism is developed to assist temporal adjacent maps' learning, enhancing discriminability. Finally, extensive experiments reveal the superiority of DTAM over existing methods on three challenging video moment retrieval benchmarks, while revealing the utilities of proposed components.

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

# A APPENDIX

## A.1 STRUCTURE OF MOMENT-AWARE MECHANISM

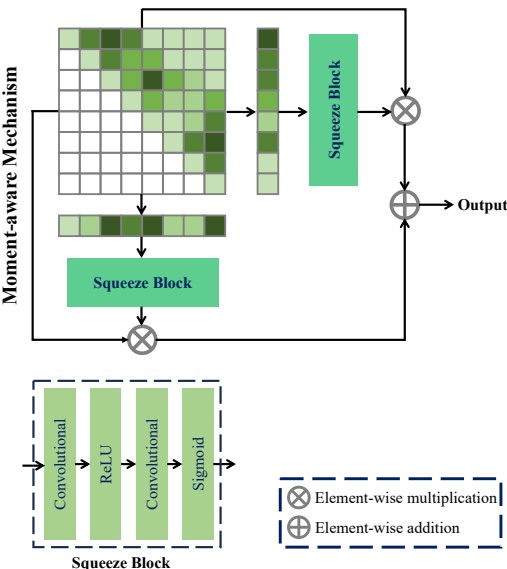

Figure 5: Illustration of the proposed moment-aware mechanism.

