# OpenReview forum: "Learning Semantic-Enhanced Dual Temporal Adjacent Maps for Video Moment Retrieval"
_ICLR.cc/2025/Conference — Submitted to ICLR 2025_

### Official Review · Reviewer_HBE6 · 2024-10-29

**Soundness:** 3
**Presentation:** 3
**Contribution:** 3
**Rating:** 6
**Confidence:** 3

**Summary:**

This paper introduces an approach, Dual Temporal Adjacent Maps (DTAM), for enhancing video moment retrieval. DTAM separates visual appearance and semantic information, addressing issues in current methods that struggle to distinguish similar-looking moments with different meanings. DTAM uses two branches to encode visual and semantic features, with the appearance branch feeding signals to the semantic branch to improve differentiation. Additionally, a moment-aware mechanism is developed to optimize the model’s attention to relevant moments. Experiments on three video retrieval benchmarks demonstrate DTAM’s superior performance, highlighting its effectiveness in capturing complex temporal and semantic dependencies.

**Strengths:**

1. DTAM’s approach of separating visual appearance and semantic information addresses a key challenge in video retrieval, allowing the model to distinguish moments that look similar but have different meanings.
2. The moment-aware mechanism in DTAM enhances the model’s sensitivity to specific moments by dynamically focusing on relevant segments.
3. The paper presents extensive experiments on three challenging benchmarks, where DTAM consistently outperforms state-of-the-art methods.

**Weaknesses:**

While DTAM achieves impressive retrieval accuracy, its dual-branch structure and moment-aware mechanism may increase computational demands. This complexity could limit its scalability and efficiency when applied to large-scale video datasets.

**Questions:**

Given DTAM’s dual-branch structure and additional moment-aware mechanism, which add complexity, what specific optimizations or architectural choices contribute to its minimal increase in inference time compared to simpler models like 2D-TAN?

---

> ### Author Response · Authors · 2024-11-20
>
> $\textbf{Response to question 1:}$
> Thanks for your valuable suggestion. Given that our dual structure processes the input in a parallel manner, it does not impose any significant additional computational burden. Furthermore, we have compared the time overhead of various model variants in Table 4(d). Notably, the complete DATM incurs only a negligible increase in time compared to 2D-TAN, while achieving a substantial improvement in retrieval accuracy. Additionally, during the training phase, the dual structure (APB + SEB) takes approximately 1.6 times longer than the variant that solely utilizes the appearance branch (APB). However, during inference, the dual structure incurs negligible additional overhead.
>
> | Model      | Training | Inference    | R@1 tIoU=0.5 |
> | :---        |    :----:   |    :----:   |          ---: |
> | 2D-TAN      | 63ms       | 23ms  | 44.51 |
> |  MS-2D-TAN   | 110ms        | 35ms      | 46.16|
> |||||
> |  APB      |  47ms |  20ms |   47.01 |
> |  APB+MAM | 49ms |  22ms |   48.81
> |  APB+SEB | 71ms |  25ms |  50.99
> |  APB+SEB+MAM  |  74ms |  31ms |  52.44

---

> > ### Comment · Reviewer_HBE6 · 2024-11-26
> >
> > I have read the author's response, and I maintain my rating and confidence.

---

### Official Review · Reviewer_4XrJ · 2024-10-29

**Soundness:** 3
**Presentation:** 3
**Contribution:** 3
**Rating:** 6
**Confidence:** 4

**Summary:**

The paper proposes a novel model for video moment retrieval, DTAM, which introduces dual temporal adjacent maps to enhance retrieval accuracy. It designs a semantic-aware contrastive loss that clusters features for the same query while distancing those for different queries, and incorporates a moment-aware mechanism to further strengthen the temporal adjacent maps. Through systematic experiments on three benchmark datasets and ablation studies, the paper thoroughly analyzes and validates the contributions of each module.

**Strengths:**

This article has advantages in the following ways:
1.  The paper introduces the Dual Temporal Adjacent Maps (DTAM) model, which decouples visual and semantic information for video moment retrieval. This design allows the model to effectively distinguish between moments that are visually similar but semantically different, addressing the limitations of traditional methods that couple visual and semantic features.
2.  Innovative Structure Design: The paper introduces the Dual Temporal Adjacent Maps (DTAM) model, which decouples visual and semantic information for video moment retrieval. This design allows the model to effectively distinguish between moments that are visually similar but semantically different, addressing the limitations of traditional methods that couple visual and semantic features.
3.  The introduction of the moment-aware mechanism allows the model to dynamically adjust the importance of video segments. This mechanism strengthens the representational power of the temporal adjacent maps in the video moment retrieval task, enabling better capture and modeling of temporal relationships in videos.

**Weaknesses:**

I think this is a convincing paper. The research questions are all reasonable. However, I believe that some improvements can be made.
1. The formatting of the tables and images on pages 6 and 7 of the paper is not aesthetically pleasing. Could you consider realigning and reformatting them?  For example, the interval between table 1 and table 2 should be widened.
2. The Introduction section in Chapter 1 states that "To this end, we propose a semantic-enhanced Dual Temporal Adjacent Maps (DTAM) for effective video grounding..." uses “video grounding”, but “video moment retrieval” used in subsequent articles. It is recommended to unify the entire text.  Video Moment Retrieval is recommended, as this is also the usage of most articles.
3.  It is unclear what knowledge the two branches have actually learned. The paper suggests that one branch focuses on appearance and the other on semantics, but this seems to be a subjective interpretation.  The paper lacks explicit supervisory signals to ensure that each branch focuses on either visual or semantic features, and there is no interpretability analysis or ablation studies to demonstrate that the branches have indeed learned their claimed distinct features.  I recommend that the authors address this issue in the manuscript. You can visualize the feature map and illustrate its results

**Questions:**

Clarification on Branch Specialization: The paper states that one branch focuses on visual appearance while the other emphasizes semantic information. However, there is no explicit supervisory signal in the model to ensure that each branch indeed specializes in its respective area. Could the authors provide clarification on how they ensure this specialization during training?

---

> ### Author Response · Authors · 2024-11-20
>
> $\textbf{Response to question 1:}$
> Thanks for your valuable suggestion. We will realign and reformate tables on pages 6 and 7, We will proceed to realign and reformat the tables located on pages 6 and 7, incorporating your feedback to enhance their overall appearance.
>
> $\textbf{Response to question 2:}$
> Thanks for your insightful suggestion. We will adhere to your recommendation and consistently use the expression "video moment retrieval" throughout our work.
>
> $\textbf{Response to question 3:}$
> Thanks for your insightful suggestion. The appearance branch is tasked with modeling visual appearance information, while the semantic branch is designed to distinguish instances with similar appearances by encoding semantic cues. To accomplish this goal, we incorporate predictions from the appearance branch into the semantic branch through distillation, aiding the DTAM in distinguishing features with identical visual content but differing semantics. This is achieved using a well-crafted semantic-aware contrastive loss. Besides, we explore various distillation strategies, including the use of dynamic thresholds, fixed thresholds, and a pre-trained appearance branch. Detailed experimental results showcasing the effectiveness of our approach are presented in Figure 3(d).
>
> As you pointed out, this is indeed our subjective interpretation of the original version. Therefore, in addition to investigating various distillation methods for validation, I have also crafted visualizations of some complex examples to clarify the underlying idea. Notably, we have incorporated a visual heat map as your insightful suggestion. However, please note that due to the constraints of the rebuttal stage, we cannot upload images in the modified version. Thanks very much again.

---

> > ### Comment · Reviewer_4XrJ · 2024-11-23
> >
> > Thank you for your feedback. I look forward to the revised manuscript showing the differences between visual features and semantic features.

---

### Official Review · Reviewer_wft7 · 2024-10-31

**Soundness:** 3
**Presentation:** 2
**Contribution:** 2
**Rating:** 3
**Confidence:** 5

**Summary:**

This paper addresses the challenge of retrieving specific moments from untrimmed videos using text descriptions by proposing the semantic-enhanced Dual Temporal Adjacent Maps (DTAM) framework. DTAM consists of two branches: one for encoding visual appearance and the other for encoding semantic knowledge from video clips. The visual appearance branch distills information into the semantic branch, enabling DTAM to distinguish features with identical visual content but differing semantics through a semantic-aware contrastive loss. Furthermore, a moment-aware mechanism is introduced to improve the learning of temporal adjacent maps for enhanced video grounding. Extensive experiments demonstrate that DTAM outperforms existing state-of-the-art methods across three benchmarks: TACoS, Charades-STA, and ActivityNet Captions.

**Strengths:**

- This paper models temporal dependencies between moments in a decoupled appearance-semantic manner, enabling differentiation between instances that have similar appearances but different semantics.
- Experiments demonstrate its effectiveness.

**Weaknesses:**

- The paper presents challenges in aligning descriptions, such as stating that "the semantic branch absorbs semantic knowledge from the appearance branch via the semantic-aware contrastive loss." However, Eqn. (7) still uses the embedding from the appearance branch; how is knowledge distillation reflected in this? Additionally, why is the appearance branch used for positive and negative sample classification instead of the semantic branch?
- In Section 3.5, the paper introduces a moment-aware mechanism aimed at enhancing the semantics of temporal adjacent maps by emphasizing the importance of each moment. However, it appears that this moment-aware mechanism is only applied to the semantic branch.
- The main contribution of this paper lies in the introduction of the contrastive loss function and the probability constraints on the start and end points in the maps. However, since contrastive loss is a common approach in cross-modal retrieval for enforcing semantic alignment, the only significant contribution appears to be the introduction of the start and end point probability values, which seems relatively weak for ICLR.
- In the experiments, specifically in the ablation study, what do the symbols APB, MAM, and SEB refer to?
- Lacking comparison with the work "SnAG: Scalable and accurate video grounding" CVPR 2024.

**Questions:**

- The paper presents challenges in aligning descriptions, such as stating that "the semantic branch absorbs semantic knowledge from the appearance branch via the semantic-aware contrastive loss." However, Eqn. (7) still uses the embedding from the appearance branch; how is knowledge distillation reflected in this? Additionally, why is the appearance branch used for positive and negative sample classification instead of the semantic branch?
- In Section 3.5, the paper introduces a moment-aware mechanism aimed at enhancing the semantics of temporal adjacent maps by emphasizing the importance of each moment. However, it appears that this moment-aware mechanism is only applied to the semantic branch.
- The main contribution of this paper lies in the introduction of the contrastive loss function and the probability constraints on the start and end points in the maps. However, since contrastive loss is a common approach in cross-modal retrieval for enforcing semantic alignment, the only significant contribution appears to be the introduction of the start and end point probability values, which seems relatively weak for ICLR.
- In the experiments, specifically in the ablation study, what do the symbols APB, MAM, and SEB refer to?
- Lacking comparison with the work "SnAG: Scalable and accurate video grounding" CVPR 2024.

---

> ### Author Response · Authors · 2024-11-13
>
> 1. Thanks for your comments. It appears that some misunderstandings have arisen. Equation (7) utilizes the feature embedding derived from the semantic branch, as opposed to the appearance branch, as depicted in Figure 2. Additionally, the algorithm now incorporates knowledge distillation by processing the output confidence of branch A (representing the acquired knowledge) to ascertain high-confidence intervals for both positive and negative instances.
> 2. In the original paper, the proposed moment-aware mechanism is used in both the appearance and semantic branches, as described in L266 and Figure 2.
> 3. Thanks for your comments. The primary novelty of our paper does not lie in contrastive learning. Instead, we identified that the integration of semantic and appearance features in the video grounding task leads to suboptimal performance. In response, we introduced a framework that decouples semantics and appearance, and leverages knowledge distillation to transfer the learned knowledge from the appearance branch to the semantic branch.
> 4.  In the original paper, we have provided explanations for the symbols APB, MAM, and SEB in Table 4 for clarity.
> 5. Thanks for your comments. We have compared the proposed DTAM with SnAG in Table 2 for the ActivityNet-Captions Dataset using C3D features.

---

### Official Review · Reviewer_ygoQ · 2024-11-02

**Soundness:** 2
**Presentation:** 3
**Contribution:** 2
**Rating:** 3
**Confidence:** 4

**Summary:**

This paper studies temporal dependence schemes that decouple content and semantic information, establishing semantic-enhanced Dual Temporal Adjacent Maps for video moment retrieval, conferred as DTAM. Specifically, DTAM designs two branches to encode visual appearance and semantic knowledge from video clips respectively, where knowledge from the appearance branch is distilled into the semantic branch to help DTAM distinguish features with the same visual content but different semantics with a well-designed semantic-aware contrastive loss. Besides, a moment-aware mechanism is also developed to assist temporal adjacent maps' learning for better video grounding. Finally, extensive experimental results and analysis demonstrate the superiority of the proposed DTAM over existing state-of-the-art approaches on three challenging video moment retrieval benchmarks, i.e., TACoS, CharadesSTA, and ActivityNet Captions.

**Strengths:**

1. This paper tries to address the sophisticated temporal dependency among moments, which leads to an inability to encode temporal dependencies between moments that are crucial for moment retrieval. Therefore, it proposes semantic-enhanced Dual Temporal Adjacent Maps (DTAM) for effective video grounding.
2. The proposed DTAM achieves satisfactory performance on three public datasets.

**Weaknesses:**

0. The novelty of this paper is somewhat limited. Temporal Adjacent Map strategy was proposed in AAAI 2019 and became a popular trick for video moment retrieval. The proposed Dual Temporal Adjacent Maps method seems to incrementally modify the previous work, without any substantive theoretical analysis. Besides, the contrastive learning strategy is a normal and commonly-sued tricks.

1. Why don't the authors leverage the pre-trained semantic space provided by popular VLMs for feature alignment? Such visual and textual features from VLMs offer rich semantic knowledge and, compared to I3D/C3D and LSTM features, provide better semantic alignment. It is important for matching appropriate video clips with queries.

2. Moreover, the paper states, "Previous video moment retrieval method ignore the flexibility and complexity of moment description, resulting in an inability to encode temporal dependencies between moments that are crucial for moment retrieval." The viewpoint needs to be supported in the experiments or with reference to other published results. Specifically, how are complex queries (e.g., those containing "again" as mentioned before) defined, and for samples containing these complex queries, what are the evaluation results of the current methods compared to the proposed DTAM?

3. What distinguishes the appearance temporal adjacent map from the semantic temporal adjacent map, and what are their respective roles? And how is temporal and semantic decoupling accomplished?

4. A previous study [1] suggests that current video moment retrieval approaches, influenced by dataset distribution, lead models to learn dataset-specific biases. What are the similarities and differences between the motivations of this work and those of the proposed DTAM? Additionally, it would be beneficial to evaluate the model's performance on Charades-CD and ActivityNet-CD.

[1] Hao J, Sun H, Ren P, et al. Can shuffling video benefit temporal bias problem: A novel training framework for temporal grounding[C]//European Conference on Computer Vision. Cham: Springer Nature Switzerland, 2022: 130-147.

**Questions:**

Please refer to the weakness part, especially the novelty issue.

---

> ### Author Response · Authors · 2024-11-13
>
> $\textit{Explaination}$ [0]: Temporal Adjacent Map (TAM), proposed in AAAI 2019, is an efficient paradigm for processing cross-modal video grounding. In this paper, we follow this paradigm and develop a dual TAM to decouple semantic and appearance features, which is our motivation that interested video clips have similar visual appearance but different semantics. For this reason, the goal of this paper is to explicitly encode semantic and appearance information in a decoupled manner, facilitating differentiation between instances with similar appearance but different semantics. Besides, contrastive learning is only a tool utilized to distinguish various samples that guarantee semantic consistency with a knowledge distillation from the appearance branch into the semantic branch. Besides, we also proposed a novel moment-aware mechanism to augment vanilla temporal adjacent maps so that they can perceive the importance of each moment.
>
> $\textit{Explaination}$ [1]: Thanks for your comments. Although the features extracted by LLM have stronger expressive power, feature optimization is not our focus. The core issue of this paper is that interested video clips have similar visual appearance but different semantics, so we proposed a novel decoupled temporal adjacent map framework to address it. Furthermore, currently popular methods basically use C3D or I3D features. To this end, for a fair comparison, I also used them to verify the effectiveness of the model. Nevertheless, we will also explore algorithm paradigms based on large model features in subsequent work based on the reviewers' valuable suggestions.
>
> $\textit{Explaination}$ [2]: Thanks for your insightful suggestion. The literature [1] elucidates and validates the phenomenon that "previous video moment retrieval methods overlook the flexibility and complexity of moment description, leading to an inability to encode the temporal dependencies between moments that are pivotal for moment retrieval." In the revised version, we will incorporate citations in accordance with your valuable feedback.
> Moreover, certain complex queries encompass similar content but possess greater semantic differences, as exemplified in Figure 1(c). To validate this, we conducted ablation studies presented in Table 4, exploring the impact on semantic modeling. Furthermore, we provide visual examples in Figure 4, comparing different variants of the diversion model. These results demonstrate that the proposed Dual Temporal Adjacent Map (DTAM) can effectively handle such complex queries.
>
> [1] S. Zhang, H. Peng, J. Fu, and J. Luo. "Learning 2D Temporal Adjacent Networks for Moment Localization with Natural Language" in AAAI 2020.
>
> $\textit{Explaination}$ [3]: Thanks for your comments. The motivation behind our work stems from the observation that interesting video clips often exhibit similar visual appearances yet possess distinct semantics. To address this, we explicitly encode appearance and semantics in a decoupled manner, enabling us to differentiate between instances that share similar visual appearances but possess different semantics. Specifically, the appearance branch is tasked with modeling visual appearance information, while the semantic branch is designed to distinguish instances with similar appearances by encoding semantic cues.
> To accomplish this goal, we incorporate predictions from the appearance branch into the semantic branch through distillation, aiding the Dual Temporal Adjacent Map (DTAM) in distinguishing features with identical visual content but differing semantics. This is achieved using a well-crafted semantic-aware contrastive loss. Additionally, we explore various distillation strategies, including the use of dynamic thresholds, fixed thresholds, and a pre-trained appearance branch. Detailed experimental results showcasing the effectiveness of our approach are presented in Figure 3(d).
>
> $\textit{Explaination} $[4]: Thanks for your valuable suggestions. The underlying motivation of [1] stems from the fact that current models learn dataset-specific knowledge, largely influenced by the distribution of the dataset. Conversely, our motivation is to disentangle visual appearance from semantic information. Both works share a common ground in mining the pivotal factors that impact retrieval performance from a data-centric perspective. However, while [1] endeavors to learn the biases between different data, our method identifies visually similar examples within the data that possess vastly different semantics. To address this, we propose a novel semantic-enhanced dual temporal adjacent map, which models temporal dependencies between moments in a manner that decouples appearance from semantics.

---

> > ### Comment · Reviewer_ygoQ · 2024-11-26
> >
> > Thank you for the author's response. However, I believe the issues related to innovation and theoretical contribution have not yet been addressed, so I am maintaining my original score.

---

### Meta-Review · Area_Chair_6CYq · 2024-12-19

**Metareview:**

This paper was reviewed by four experts in the field. The final ratings are 6,6,3,3. This paper proposes a dual temporal adjacent maps for video moment localization.

Reviewers generally agree that this paper addresses an interesting problem. The proposed method shows reasonable performance improvement. Reviewers also raises several concerns. The rebuttal addressed some of the concerns. But some of the concerns are still remaining, such as the novelty, the fairly subjective interpretation of appearance vs semantic branches, etc. Given these concerns, the decision is that the paper is not ready for ICLR in its current form. Authors are encouraged to take into account of reviewers' comment and revise the paper for resubmission.

**Additional Comments On Reviewer Discussion:**

The rebuttal addressed some of the reviewers' concerns. But some concerns are still remaining.

---

### Decision · Program_Chairs · 2025-01-22

Reject